# Relations of Bedtime Mobile Phone Use to Cognitive Functioning, Academic Performance, and Sleep Quality in Undergraduate Students

**DOI:** 10.3390/ijerph17197131

**Published:** 2020-09-29

**Authors:** Darnisha Ragupathi, Normala Ibrahim, Kit-Aun Tan, Beatrice Ng Andrew

**Affiliations:** Department of Psychiatry, Faculty of Medicine and Health Sciences, Universiti Putra Malaysia, Serdang 43400, Malaysia; normala_ib@upm.edu.my (N.I.); tanka@upm.edu.my (K.-A.T.); beatrice_andrew@upm.edu.my (B.N.A.)

**Keywords:** bedtime mobile phone use, sleep quality, academic performance, cognitive functioning, young adults

## Abstract

The present cross-sectional study examined the relations of bedtime mobile phone use to cognitive functioning, academic performance, and sleep quality in a sample of undergraduate students. Three hundred eighty-five undergraduate students completed a self-administered questionnaire containing sociodemographic variables, bedtime mobile phone use, the Pittsburgh Sleep Quality Index, and the Cambridge Neuropsychological Test Automated Battery (attention and verbal memory). At bivariate level, increased scores in bedtime mobile phone use were significantly correlated with decreased scores in academic performance and sleep quality. Our multivariate findings show that increased scores in bedtime mobile phone use uniquely predicted decreased scores in academic performance and sleep quality, while controlling for gender, age, and ethnicity. Further untangling the relations of bedtime mobile phone use to academic performance and sleep quality may prove complex. Future studies with longitudinal data are needed to examine the bidirectional effect that bedtime mobile phone use may have on academic performance and sleep quality.

## 1. Introduction

In recent years, the use of mobile phones has proliferated and brought with it a great influence on human communication through its rapid advancement [1,2]. Mobile phones are well received by users for two reasons: (a) multifunctionality: mobile phones are equipped with features such as an alarm clock, music player, games, internet, short message service (SMS), and video call, and (b) availability: mobile phones have become ubiquitous, enabling individuals to connect instantly and to engage in real-time communication at any time [3,4,5]. For these reasons, mobile phone is said to have the highest percentage of availability in the bedroom at night compared to other technological devices [6].

### 1.1. Mobile Phone Use in Young Adults

Mobile phones have attracted users of all ages. However, mobile phones are known to be vastly popular among young adults. A survey by the Malaysian Communication and Multimedia Commission [7] revealed that mobile phone use is dominated by young adults compared to teenagers, older adults, and senior citizens. To this end, mobile phones are particularly well received by university students for two reasons. First, mobile phones have become vital among university students as they accommodate their need of information-seeking and provide educational assistance. Second, mobile phone applications provide university students with access to instant communication and social media (e.g., WhatsApp, Facebook, and Instagram) at any point of the day [8]. 

### 1.2. Bedtime Mobile Phone Use

Bedtime mobile phone use is defined as constant use of the mobile phone while in bed, a few nights per week within the hour before sleep, and hesitancy to switch off the device at night [9,10,11,12]. In addition, bedtime mobile phone use can also be manifested through a frequency of calls and text messages made and/or received after lights out [13]. The use of a mobile phone specifically at night could serve many distinct purposes. However, it has raised concerns, as the period of engagement during the hours before sleep is alarming [9,10], with about 68% of young adults reporting a regular usage of such a device in bed [14]. 

### 1.3. Impacts of Bedtime Mobile Phone Use on Cognitive Functioning, Sleep Quality, and Academic Achievement

Among various domains of cognitive functioning, research has found attention and memory to be most closely associated with mobile phone use. The exposure to light-emitting devices at night could have a negative impact on attention and verbal memory [15,16]. The use of mobile phone at night leads to loss of mental attention and decline in other areas of cognitive performance, such as memory, due to feeling of fatigue and tiredness [17]. 

Bedtime mobile phone use could negatively affect academic performance [18]. The development of the educational system in accordance with technological advancements has led to an increase in mobile phone use for academic purposes. The traditional method of studying and reading books has been substituted with an alternative method of instant fingertip access such as the use of electronic books, online journals, and internet-based studying applications such as MedCalc, Prognosis, and Lexicomp [2,19]. This has contributed to a change in the pattern of mobile phone use, during and after school hours. Students have higher levels of engagement with this device at night for studying purposes and subsequently engaging in free night calls, chatting, instant messaging, and social media that are reportedly influencing their academic performance [2]. 

Researchers have conducted a great deal of studies relating mobile phone use during bedtime and sleep quality in children and adolescents [6,9,20]. Mobile phones are commonly kept on the bed or beside the bed while sleeping, resulting in difficulty falling asleep due to constant awakening by notifications and trouble switching off the phone during bedtime [18]. Moderate use of the device at night prior to bedtime could contribute to poor sleep quality [10]. Exposure to the light emitted by the mobile phone device causes melatonin suppression and increases the rate of stimulation in the circadian system, leading to sleep issues [20,21]. Individuals with sleep issues such as insomnia might have high engagement of bedtime mobile phone use to pass the time [22]. 

### 1.4. Control Variables

Based on the existing findings, sociodemographic variables such as gender [13], age [23], and ethnicity [1,24] can play a role in predicting cognitive functioning, academic performance, and sleep quality. Control variables are included in the present study to account for the effects of additional variables that may influence outcome variables. 

### 1.5. The Present Study

The current study aimed to address at least three research gaps. First, young adults have the highest level of mobile phone engagement among all demographic groups [1,25], yet research on mobile phone use has been primarily limited to adolescents and children [14,25,26]. In the present study, we identified undergraduate students as our study population. Second, in the absence of a reliable and valid measurement, it is difficult if not impossible to assess bedtime mobile phone use among young adults. Informed by Exelmans and Bulck [13], we developed a scale for assessing bedtime mobile phone use in the present sample. Third, numerous studies have examined the relationship between bedtime mobile phone use and isolated domains of cognitive functioning [16,23]. Few studies have simultaneously examined the relationships between bedtime mobile phone use and multiple domains of cognitive functioning. In response, we included attention and verbal memory as domains of cognitive functioning. These taken together, the primary purpose of the present study was to examine the relations of bedtime mobile phone use to cognitive functioning, academic performance, and sleep quality in a sample of undergraduate students. We hypothesized that bedtime mobile phone use would improve the prediction of cognitive functioning (attention and verbal memory), academic performance, and sleep quality beyond that provided by gender, age, and ethnicity.

## 2. Methods

### 2.1. Participants

Three hundred and eighty-five undergraduate students (73% female and 27% male) across eight different faculties at Universiti Putra Malaysia participated in this study. Participants were selected via the multistage stratified sampling method. The mean age of the sample was 21 years (*SD* = 1.11), ranging from 20–24 years old. Ethnicity breakdown for the sample revealed that 73.8% of the participants were Bumiputera and 26.2% were non-Bumiputera. 

The sample size was calculated based on Lemeshow’s (1990) recommendations [27]. A sample of 385 participants was needed in the present study. Exclusion criteria for participation in the present study were students with a history of sleep medication in the past month which could affect the outcome of the study and absentees on the day of data collection. Inclusion criteria for participation in the present study were the following: Years 1 and 2 Malaysian undergraduate students aged between 20–24 years old, availability of GPA as a means of academic performance, possession of a mobile phone, and signed informed consent. 

### 2.2. Procedure

Data collection was carried out in a closed room to ensure that there were no external distractions. Participants were briefed about the study objective and were given an informed consent form. After obtaining informed consent, participants were requested to fill in a self-administered questionnaire on sociodemographics, bedtime mobile phone use, and sleep quality. In the self-administered questionnaire, we asked participants to provide their matrix number to gain access to their GPA. Information on GPA score was obtained from the academic department with permission from the Head of Academic Governance Division, Universiti Putra Malaysia. Participants were given an iPad with an automated installation of the Cambridge Neuropsychological Test Automated Battery (CANTAB) to complete a series of cognitive tests. The self-administered questionnaire was distributed first, followed by the CANTAB assessment to prevent any cognitive interference. 

### 2.3. Consent

We sought ethical approval by Ethics Committee for Research involving Human Subjects, Universiti Putra Malaysia prior to data collection. Permission and approval from each participating faculty was also obtained. Participation was strictly voluntary, and informed consent was obtained from each participant prior to data collection. 

### 2.4. Measures

#### 2.4.1. Sociodemographics

Participants’ information on sociodemographics such as gender, age, and ethnicity were obtained.

#### 2.4.2. Cognitive Functioning

The CANTAB has varying test measures that cover cognitive functioning domains such as attention, memory, executive function, decision making, and social cognition [28]. Due to the focus of the present study, we only administered Rapid Visual Information Processing (RVP) to assess attention and Verbal Recognition Memory (VRM) to assess verbal memory.

##### RVP

We used RVP to assess participants’ attention. In the present study, participants took about seven minutes to complete the test. During test administration, participants were represented with a white box displayed at the center of the screen. Digits ranging from 2–9 appeared in a pseudo-random order at a rate of 100 digits per minute inside the white box. Participants were required to detect the sequence (e.g., two–four–six, three–five–seven, four–six–eight). They then responded to the target sequence by selecting the center button of the displayed screen as quickly as possible. Possible scores range from 0–1. Higher scores indicate better attention [29]. 

##### VRM

We used the VRM Immediate Recognition and the VRM Delayed Recognition tests to assess participants’ verbal memory in the present study. Participants took about ten minutes to complete the tests. For the VRM Immediate Recognition test, participants were presented with a sequence of words on the screen one at a time. They were requested to recall the words. For the VRM Delayed Recognition tests, participants were presented with two words (one was from the original list and the other one was a distractor). They were asked to choose the words that they had seen before. For both tests, participants received scores for the correct response. Possible scores range from 0–36 for each test. Individual scores on the VRM Immediate Recognition and the VRM Delayed Recognition tests were summed to obtain a total score on verbal memory. Higher scores indicate better verbal memory [30]. 

#### 2.4.3. Academic Performance

Participants’ GPA was obtained from the academic department of Universiti Putra Malaysia. The alphabetic grade was determined based on the mark obtained by undergraduate students for a specific course. No personal identifiers were obtained. Higher GPA indicates better academic performance. 

#### 2.4.4. Sleep Quality

The 19-item Pittsburgh Sleep Quality Index (PSQI) was used to assess sleep quality [31]. A global score was produced by summing up all PSQI items [32]. Possible scores range from 0–21. Cronbach alpha was 0.75 for the PSQI in the present study.

#### 2.4.5. Bedtime Mobile Phone Use Scale

Based on Exelmans and Bulck [13], we developed a 6-item scale to measure bedtime mobile phone use in the present study (Appendix A). Permission to modify the scale was obtained from the original scale developers. Participants were asked to indicate the frequency at which they engaged in bedtime mobile phone use during the last 30 days. As with Exelmans and Bulck [13], all items are scaled on 1 (*never*), 2 (*one to three times a month*), 3 (*once a week*), and 4 (*several times a week*), and 5 (*everyday*). A sample item reads: “How frequently do you send text messages after lights out?”. A total score was calculated by summing up the 6 items. Possible scores range from 6–30. As far as the psychometric properties of the scale are concerned, we first calculated the I-CVI (item content validity index for items) and the S-CVI (content validity index for scales) to establish content validity [33] and then obtained Cronbach alpha to establish internal consistency. Both the I-CVI and the SCI indices were 1, thereby establishing content validity. Cronbach alpha values of above 0.70 are referred to as acceptable and above 0.80 as high [34]. Cronbach alpha for the Bedtime Mobile Phone Use Scale (BMPUS) in the present study was high (*α* = 0.89). Preliminary evidence of validity and reliability was obtained. The BMPUS was found to be a reliable and valid tool for assessing bedtime mobile phone in the present study.

### 2.5. Data Analytic Plan

Data were checked for missing and out of range values. Normality of the present study and univariate outliers were assessed through measure of skewness and kurtosis following Kline’s recommendations [34]. We performed Pearson’s *r* correlation tests for bivariate analyses. As for the multivariate analysis, we performed a series of hierarchical multiple regression analysis with cognitive functioning, academic performance, and sleep quality as dependent variables. Sociodemographic variables such as gender, age, and ethnicity were included in the analyses as control variables. Gender, age, and ethnicity were entered in Step 1, followed by bedtime mobile phone use in Step 2. Assumptions of multicollinearity were tested based on tolerance (>2) and variance inflation factor (VIF) (<10) values [35]. Multivariate outliers were assessed with the Mahalanobis distance using the regression function. The acceptable range of a critical χ^2^ value with a degree of freedom of three was 16.27 [35]. We used the Statistical Package for Social Sciences (SPSS)^®^ Version 25 to perform all statistical analyses. All statistical significance levels were set at *p* < 0.05.

## 3. Results

### 3.1. Descriptive Properties of Study Variables

Table 1 presents the descriptive properties of variables. The normality assumptions were achieved. 

### 3.2. Correlations among Study Variables 

Table 2 presents Pearson’s *r* correlations for the study variables. Ethnicity was significantly and positively correlated with cognitive functioning variables (attention and verbal memory) and academic performance. Age and ethnicity were significantly and negatively correlated with sleep quality. Bedtime mobile phone use was significantly and negatively correlated with academic performance but was significantly and positively correlated with sleep quality.

### 3.3. Hierarchical Regression Analyses

Table 3 shows the results of hierarchal regression analyses with cognitive functioning, academic performance, and sleep quality as dependent variables, while accounting for gender, age, and ethnicity. In the present sample, no violation of multicollinearity assumption was detected as all tolerance and VIF values were within an acceptable range (see Table 3). Multivariate outliers were assessed with Mahalanobis distance using the regression function. The critical chi-square (χ^2^) value with a degree of freedom of three was 16.27, suggesting no violation of multivariate outlier.

#### 3.3.1. Predicting Cognitive Functioning

In Step 1, gender, age, and ethnicity were not found to be significant predictors for attention and verbal memory. In Step 2, bedtime mobile phone was also not found to be predictive of attention and verbal memory.

#### 3.3.2. Predicting Academic Performance

As shown in Step 1, age and ethnicity were found to be significant predictors for academic performance. In Step 2, addition of bedtime mobile phone use made a significant contribution in predicting academic performance, accounting for an additional 1% of the variance in academic performance. Age and ethnicity retained their statistical significance.

#### 3.3.3. Predicting Sleep Quality

In Step 1, age and ethnicity were found to be significant predictors for sleep quality. In Step 2, addition of bedtime mobile phone use made a significant contribution in predicting sleep quality, accounting for an additional 3% of the variance in sleep quality. Only age retained its statistical significance.

## 4. Discussion

The present study examined the relations of bedtime mobile phone use to cognitive functioning, academic performance, and sleep quality in an undergraduate sample. At bivariate level, increased scores in bedtime mobile phone use were significantly correlated with decreased scores in academic performance and sleep quality. The findings from hierarchical regression analyses show that addition of bedtime mobile use significantly explained and increased variance in each outcome. However, coupled with sociodemographic variables, bedtime mobile phone use collectively predicted 10% of variance in academic performance and 6% of variance in sleep quality, suggesting that the remaining explanatory power in both outcome variables could be due to emotional factors such as mood [8,20,36] that needs to be fully studied in future research.

No significant relationship between bedtime mobile phone use and cognitive functioning was reported. In particular, bedtime mobile phone use was not significantly correlated with attention and verbal memory. This finding concurs with Schoeni et al. [23]. It is possible that selection bias could be present, whereby individuals who prefer to use mobile phone at night are those with higher cognitive functioning [23]. Cognitive functioning is a multidimensional construct. However, only participants’ attention and verbal memory were assessed in the present study. Inclusion of these two cognitive functioning variables alone may not be sufficient to assess its relationship with bedtime mobile phone use [37].

Our findings show that high bedtime mobile phone use was associated with low academic performance. These findings were in line with those of previous research [2,18,38,39,40]. Advancements of technology have altered the traditional methods of studying, resulting in higher engagement with technological devices for educational purposes [2,19]. This could lead to higher engagement with bedtime mobile phone use to seek information or assist with learning needs after school hours. Subsequently, individuals have a higher tendency of being absent or late for classes owing to feeling of sleepiness [18].

A global PSQI score of >5, as suggested by Buysse et al. [31], is indicative of poor sleep quality. In the present sample, 85.2% of the participants rated their sleep quality as poor. This finding is in line with previous studies examining the relation between mobile phone use and sleep quality [13,14,41]. In particular, our findings corroborate previously published results that found high bedtime mobile phone use was associated with low sleep quality [13,14,41,42]. It is now a common trend that individuals keep their mobile phone on or close to their bed during bedtime [18]. In this regard, constant notifications keep them awake resulting in an increased mobile phone use during bedtime. Using the mobile phone during bedtime for text-messaging or calling may induce cognitive or emotional arousal, which is said to be the cause of poor sleep quality [21,22,43].

The present study’s findings provide crucial information to promote mobile phone etiquette (mobiquette) in university students. Health education or workshops are beneficial for raising awareness about mobiquette; students should learn some simple etiquette around responsible use of mobile devices as a medium for studying and learning [44]. Sleep applications could help students to set limits on mobile phone use during bedtime. Studies examining the effectiveness of sleep applications for developing healthy sleeping habits have been documented [45]. For example, Sleep Ninja, a cognitive behavior therapy smartphone application, has been developed to help young people to improve sleep quality while setting limitations to mobile phone use during bedtime [11]. 

### Limitations and Recommendations

The present study has several limitations. First, the CANTAB offers a varying number of cognitive assessments. However, only two cognitive functioning domains, attention and verbal memory, were assessed in the present study. Other cognitive functioning domains such as executive functioning, decision making, and social cognition could be assessed in future studies to further explain the bedtime mobile phone use–cognitive functioning relationship [37]. Second, sole reliance on a self-reported questionnaire may result in response bias. In the present study, data on participants’ sleep quality and bedtime mobile phone use were obtained using the self-report method. Collecting information on bedtime mobile phone use via mobile phone log or accelerometer data from the smartphone would improve data accuracy [20,23]. Objective sleep measures such as actigraphy and polysomnogram could be used to assess sleep quality [46,47]. Third, the relation of bedtime mobile phone use to sleep quality is said to be both bidirectional and complex [25]. Future researchers should consider a longitudinal study design to examine the bidirectional relationship of bedtime mobile phone use to sleep quality. Fourth, in the present study, we focused only on mobile phones and did not collect data on other electronic devices. It is possible that the use of tablets or other electronic devices and related activities could serve as a moderator in the relationship between sleep quality, academic performance, cognitive functioning, and mobile phone use. Fifth, we did not measure mood or record any diagnoses of sleep, medical, or psychiatric disorders. Future mobile phone use research would benefit from the inclusion of measures of mood and psychiatric functioning. Last but not least, we developed the BMPUS to assess bedtime mobile phone use in the present study. Only preliminary evidence of content validity and internal consistency was obtained. The scale awaits future psychometric investigations following established requirements [48]. 

## 5. Conclusions

Bedtime mobile phone use is a relatively new topic in media studies, and this study represents the first step toward understanding the emergence of bedtime mobile phone use. We attempted to investigate the effect of bedtime mobile phone use on cognitive functioning, academic performance, and sleep quality in a sample of undergraduate students. Our findings show that higher scores in bedtime mobile phone use uniquely predicted lower scores in academic performance and sleep quality, while adjusting for the effect of gender, age, and ethnicity. Further untangling the relations of bedtime mobile phone use to academic performance and sleep quality may prove complex. Future studies with longitudinal data are needed to examine the bidirectional effect that bedtime mobile phone use may have on academic performance and sleep quality.

## Figures and Tables

**Table 1 ijerph-17-07131-t001:** Descriptive properties of study variables (*n* = 385).

Variables	*M*	*SD*	Skewness	Kurtosis
1. Attention	0.58	0.15	0.23	−0.10
2. Verbal Memory	63.41	5.26	−0.85	0.41
3. Academic Performance	3.41	0.34	−0.57	−0.20
4. Sleep Quality	7.02	2.83	0.50	0.07
5. BMPU	19.36	5.91	−0.19	−0.77

Note. BMPU = Bedtime mobile phone use.

**Table 2 ijerph-17-07131-t002:** Intercorrelations among study variables (*n* = 385).

Variables	1	2	3	4	5	6	7	8
1. Age								
2. Gender	−0.08							
3. Ethnicity	0.11 *	−0.02						
4. Attention	0.02	−0.007	0.13 *					
5. Verbal Memory	0.01	−0.05	0.11 *	0.90 ***				
6. Academic Performance	−0.10	0.06	0.20 ***	0.14 **	0.09			
7. Sleep Quality	−0.14 **	−0.007	−0.12 *	−0.09	−0.06	−0.09		
8. BMPU	0.003	0.09	−0.27 ***	−0.10	−0.07	−0.17 **	0.19 ***	

Note. Gender (1 = Male); ethnicity (1 = Bumiputra); BMPU = bedtime mobile phone use; * *p* < 0.05, ** *p* < 0.01, *** *p* < 0.001.

**Table 3 ijerph-17-07131-t003:** Hierarchical multiple regression predicting cognitive functioning (attention and verbal memory), academic performance, and sleep quality (*n* = 385).

	Attention					Verbal Memory					Academic Performance					Sleep Quality				
Predictors	*b*	*SEb*	*β*	T	VIF	*b*	*SEb*	*β*	T	VIF	*b*	*SEb*	*β*	*T*	VIF	*b*	*SEb*	*β*	*T*	VIF
Step 1																				
Gender	−0.01	0.02	−0.03	0.99	1.0	−0.58	0.60	−0.05	0.99	1.0	0.05	0.04	0.07	0.99	1.0	−0.07	0.32	−0.01	0.99	1.0
Age	0.01	0.01	0.06	0.98	1.0	0.01	0.24	0.002	0.98	1.0	−0.04	0.02	−0.12 *	0.98	1.0	−0.32	0.11	−0.13 *	0.98	1.0
Ethnicity	0.03	0.03	0.08	0.98	1.0	1.3	0.61	0.11	0.98	1.0	0.20	0.04	0.29 ***	0.98	1.0	−0.70	0.33	−0.11 *	0.98	1.0
Δ*R*^2^	0.01					0.01					0.01 ***					0.03 *				
Step 2																				
Gender	−0.01	0.02	−0.02	0.99	1.0	−0.54	0.61	−0.05	0.99	1.0	0.06	0.04	0.08	0.99	1.0	−0.17	0.32	−0.03	0.99	1.0
Age	0.01	0.01	0.06	0.98	1.0	0.02	0.24	0.003	0.98	1.0	−0.04	0.02	−1.2 *	0.98	1.0	−0.34	0.13	−1.3 **	0.98	1.0
Ethnicity	0.02	0.02	0.06	0.91	1.1	1.2	0.64	0.01	0.91	1.1	0.20	0.04	0.26 ***	0.91	1.1	−0.37	0.33	−0.06	0.91	1.1
BMPU	−0.001	0.001	−0.06	0.92	1.1	−0.03	0.05	−0.03	0.92	1.1	−0.01	0.003	−0.11 *	0.92	1.1	0.09	0.03	0.18 **	0.92	1.1
Δ*R*^2^	0.01					0.02					0.10 ***					0.06 ***				

Note. *b* = beta values; *SEb* = standard error beta values; *β* = standardized beta values; Δ*R*^2^ = *R* square change; VIF = variance inflation factor; Gender (1 = Male); ethnicity (1 = Bumiputra); BMPU = bedtime mobile phone use; * *p* < 0.05, ** *p* < 0.01, *** *p* < 0.001.

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
