# Peer review of "Relations of Bedtime Mobile Phone Use to Cognitive Functioning, Academic Performance, and Sleep Quality in Undergraduate Students"

_ijerph, 2020, doi:10.3390/ijerph17197131_

Round 1

Reviewer 1 Report

The authors examined cognitive function (attention and verbal memory), academic performance, self-report sleep quality in relation to bedtime mobile phone use. Overall, the research question is important for health and functioning among emerging young adults. The strengths of this study include 1) large sample size, 2) standardized cognitive testing procedures, and 3) naturalistic study, such that the results could provide realistic feedback and inform possibly public health interventions. There are a few things the authors may wish to consider, which, in my view, would strengthen the manuscript.

Introduction:

1. The authors reviewed the literature on how mobile use potentially affects sleep quality and cognition. As the authors stated, there is considerable evidence in this area. The introduction would be strengthened by expanding on what this study would add to the literature. Particularly, the rationale of why the opposite direction (mobile use as the dependent variable) was specifically examined would help set out research goals. 

2. Page 3, Line 125: Please clarify “over and above cognitive functioning” in hypotheses. Were the authors meant to compare the strengths of these three variables as related to bedtime cellphone use?

Methods

1. Could the authors clarify when “VRM delayed recognition” test was conducted? What’s the time interval between immediate and delayed testing? Also, were volunteers trained on cognitive tasks?

2. Could the authors provide more detail on the Bedtime Mobile Phone Use Scale, such as example questions? Was the duration of mobile phone use at bedtime asked on the scale?

Results:

1. The average PSQI score seems high. What’s the rate of poor sleeper (PSQI>5)? 

2. Were demographic variables adjusted for in regression models?

Discussion

1. It would be helpful if the authors briefly discussed sleep quality and mobile phone use of this sample, especially in comparison with other samples.

2. Total verbal memory score was not correlated with bedtime mobile phone use. Did the authors test immediate recall and delayed recall separately? Also, was the time of attention and memory tests recorded in the study? I would think the performance may be quite different at 8 am, 11 am, 2 pm, or 7 pm for the same individual.

3. Page 9 Line 269: “Exposure to light-emitting devices at night…” The cited experimental study focused on evening exposure of LED computer screen and immediate cognitive response, which is not comparable to associations between habitual cellphone use at bedtime and cognitive function in general in this study.  Also, did the authors collect data on the use of tablets or other electronic devices? If not, could the authors discuss how this might influence research findings?

4. The authors examined how sleep quality and GPA affect mobile phone use. However, limited screen time before bedtime is part of sleep hygiene that promote sleep quality. Optimal sleep quality is essential to cognitive function and academic performance. The authors may consider incorporating the reciprocal associations into the discussion. Such information is critical to inform modifiable factors among these intertwined variables. 

5. Page 10, Line 317 “Sleep trackers such as FitBit and Sleep Cycle …”. These commercial sleep trackers are not established sleep assessment instruments. Actigraph and PSG are recommended objective sleep measurements.

6. Table 2: numbers were not aligned.

Reviewer 2 Report

Dear Authors,

I find the study brings forth very important and cutting-edge findings on the correlations between cognitive ability, academic performance, etc. with bedtime mobile phone usage.

The instruments used and the results are in line with many published literature and I find that welcome.

One aspect which is clearly highlighted is the skewed gender representation/ proportion in your study sample. Its 73% women compared to 27% men. I would expect you to look at variations in the gendered nature of bedtime phone usage and its linkage to the parameters that you have considered.

Also, this gender skewness should be highlighted as a limitation in your limitations section.

Reviewer 3 Report

This study investigates how mobile phone use in bed is related to cognition, academic performance, and sleep in university students. On the surface, this is an interesting topic which has garnered much attention in recent years. However, I have several concerns with this manuscript, particularly regarding the data analysis plan. Overall, I believe the study can add to the field, however extensive editing of the manuscript is required, and possibility re-analysis of the data.

Introduction

1) The introduction is very long, and could easily be condensed and made more concise. Many of the topics covered are not required for a research paper introduction, and it results in the introduction lacking focus. While not an exhaustive list, some specific examples:

- Detailing that mobile phones gained popularity in the late 1990s is not necessary (Page 1 line 27)

- Page 1 line 29: Sentences not well constructed, not grammatically correct and many short sentences – should read something like:

“The mobile phone is well received for two reasons: 1) multi-functionality: mobile phones are equipped with features such as an alarm clock, music player, games, internet etc.; 2) availability: mobile phone use has become ubiquitous within society, enabling individuals to connect instantly (REFs).

- Including the UN definition of younger adults doesn’t feel necessary (Page 1 line 41)

- Information about university students does not add anything beyond what has already been said about the uses of mobile phones (Page 2 lines 42-46)

- Section 1.2 needs to be made much clearer. The first sentence defines bedtime mobile phone use as “constant use” (line 48), however line 53 then states that use of mobile phones in bed might be due to setting alarms – this would not constitute as bedtime phone use, based on your own definition. I would remove this sentence, as this is not the type of behaviour you are interested in.

- Your prevalence statistics are unclear – on line 56 you state “the prevalence in Bulck’s 2007 study”. This is not meaningful to the reader. If you are interested in young adults, all that needs to be stated is “Bedtime mobile phone use in young adults is highly prevalent – with XX-XX% reporting regularly using their phones in bed (REFS).” Instead of listing every study separately.

-Similarly, stating “past studies have used self-administered questionnaires to collect information” – you can just state “self-reported phone use XXXXX” – you do not actually state any results in this section, you simply state that studies have used self-administered questionnaires, and then provide the references. Why is this important? This section either needs to be revised to reflect their findings, or removed (lines 62-67). A review of previous methodologies doesn’t really seem relevant or necessary here.

- The first paragraph of section 1.3 could probably be removed (line 69-76) – as these topics are not relevant to your research question it seems tangential to mention them. Section 1.3 could simply be called Impact of Bedtime Mobile phone use on sleep and cognition, which is what you’re interested in. You also do not need to define cognition (line 77)

- Section 1.3 also lacks clarity and focus. How is bedtime phone use related to sleep and circadian rhythms? How is bedtime phone use related to cognition?

- Definitions such as defining grade point average, sleep quality etc. are all not necessary

- Detailing the difference between self-reported GPA and institution reported GPA feels unnecessary

2) Your aims and hypotheses are worded in such a way that cognition and sleep will predict phone use – aka those with higher/lower(?) cognition and those with better/poorer(?) sleep quality will use their phones in bed more. Although you do not explain in which direction you think these predictions will go – i.e. will people will better or poorer sleep use their phones more? It also seems like a strange choice to have phone use as the dependent variable. It is unclear to me why you think someone’s sleep quality or grades would predict how much they use their phones in bed. This also is not consistent with the wording of the rest of your introduction.

Method

3) Line 139: What cultural differences and background was considered exclusionary? This seems to be discriminatory without any additional information.

4) Was any screening conducted for mood disorders? It is not listed as an exclusion criteria, and no mood questionnaires are listed as a material. As mood disorders and mood symptoms will have large effects on sleep, cognition, and maybe also phone use, this is a very large confound in the current study.

5) The choice to limit your results to Processing speed and verbal memory seems curious to me. Did you collect any other measure also included in the CANTAB battery? If not, why not? I think your introduction needs to justify why these outcomes in particular are important, and what in the prior literature led you to choose these over others.

6) Your introduction states that there a number of studies have used self-reported bedtime phone use, and then states that there is a lack of validated scales – which seems surprising and confusing to me. You then developed your own scale, however do not provide any evidence that it is in fact a good measure of objective phone use. Cronbach’s alpha only assesses internal consistency – i.e. that the items relate to each other strongly – the items may still not be a true reflection of actual phone use, and so does not seem to address the issue you have raised of “un-validated scales”.

7) Same issue with data analysis section as I raised in point (2) – it seems counterintuitive that bedtime phone use has been entered as the dependent variable.

Results

8) Including mean, SD, skew and kurtosis in the correlation table is unusual and quite confusing visually, I would amend this table.

9) Table 2 is not readable – there has been a formatting error somewhere along the way

Discussion

10) The opening paragraph of the discussion should summarise all of the main findings of the study, instead of only discussing one null finding, which should be discussed in greater detail in subsequent paragraphs

11) Line 268 – The Cajochen paper shows that evening light results in immediate cognitive improvements, due to the direct alerting effect of light, and immediate suppression of melatonin, leading to immediate changes in cognition. This is not really relevant to your paper, as you did not test cognition in the evening, while they were using their phones.

12) Lines 276-282: Discussion about using their phones to study more – however did you assess this? Did you ask them why they were using their phones? Are you able to disentangle using a mobile phone to study or seek resources, versus for social media?

13) Line 291: you state that your findings show that bedtime phone use represents an emerging public health issue. Your findings do not support this at all. You found that individuals with poorer sleep were more likely to use their phones – this does not represent a public health issue per se – the poor sleep does. You also did not investigate mood and so the implication from the previous sentence regarding mood disorders is a long stretch from your results. Finally, your correlation values were very small, and only statistically significant due to your large sample size. The effect size is not clinically meaningful and the variance explained is < 5%. You need to discuss this in your discussion and do not stretch the interpretation of your results beyond this.

14) The final paragraph of page 9 (numbered 1) discusses avoiding using mobile phones before bed, etc. however this once more does not suit your analysis plan of bedtime phone use being the dependent variable. Your analysis plan SHOULD suggest that individuals improve their sleep and grades, and this will lower their bedtime phone use (which doesn’t really make sense).

15) Line 299: The line “unfavourable effect associated with increased phone use” once again implies a different analysis plan

16) Line 300 – 304: The suggestion that policy makers could bring in laws and policies regarding personal phone use in an individual’s home is a little bit outrageous. I would remove this whole section.

17) A stronger conclusion regarding what you actually found and what it means would be useful, instead of stating “contribute and broadens the body of knowledge” – in what way?

Reviewer 4 Report

  It is an important study nowadays due to the increased use of new technologies and the impact on mental health. Sleep alterations have been described previously and just fail because there is no group with neurophysiological evaluation that could correlate neuropsychological data with these data.

Author Response

Response to Reviewer #4 Comments

It is an important study nowadays due to the increased use of new technologies and the impact on mental health. Sleep alterations have been described previously and just fail because there is no group with neurophysiological evaluation that could correlate neuropsychological data with these data. 

Response: We would like to express our gratitude to your positive response to the manuscript. We sincerely hope you will find this revision acceptable.

Round 2

Reviewer 1 Report

This version of the manuscript has been improved substantially. However, I have two major comments:

1. As mentioned in my initial review, it’s less convincing to consider betime mobile phone use as a dependent variable predicted by sleep quality and cognition. Even though there may be bi-directional associations, reducing screening time is a more modifiable factor to improve sleep quality and cognitive function, rather than the opposite direction that was examined in this study. It seems that "addressing sleep and cognitive impairment so as to reduce betimes mobile phone use" has little implication for future intervention and practice.   

2. Intuitively, social and demographic variables such as age, SES may affect sleep, cognition, screen time, and their correlations. Since data was available, the authors may repeat the analysis with adjustment for these covariates and report whether results remain the same in this sample.

Reviewer 3 Report

1) The introduction has been improved substantially, however there is still a large issue with the data analysis plan and reversal of causation for the present study. Section 1.3 of the introduction clearly sets up bedtime phone use to be the independent variable, not the dependent variable. This is clear in lines 58, 59, 62, 75-81. As raised in my initial review, there is no theoretical or scientific reason provided as to why mobile phone use would be the dependent variable in the current study.

2) Section 2.4.2.1 – There is no information about what this task actually involves. Please provide some information regarding the task and what participants were actually required to do. Additionally, the abbreviations VRMIRTC and VRMDRTC are not helpful or intuitive and are rather meaningless to your audience. Please revise this section to clearly state the procedure and how the outcomes were obtained, and then give them straight forward, meaningful names such as “Immediate Recognition” and “Delayed Recognition”. Finally, is it valid to sum the two scores? While I am unfamiliar with CANTAB scoring, this seems rather unusual for a memory test, as immediate and delayed testing tap into different processes.

3) Section 2.4.2.2 - Please also provide information regarding this task and what participants were required to do.

4) Please include the mobile phone use scale as a supplementary material, seeing as this scale has never been used before.

5) Just because demographic variables are not modifiable does not mean they are not impacting on your results and data. Given you do not have an equal sex split in your sample, and you have multiple ethnic groups represented in your sample, it is entirely justified and required that you at least test whether there are any relationships between these two demographic variables and any of your outcomes, and include them as covariates if any relationships exist. The Spector and Brannick reference is not a justification to not control for covariates.

6) Table 1 is still very confusing and does not follow standard table formatting. Please consider splitting into two tables (descriptives and correlations separately).

7) Section 3.2 is still unjustified in the direction of the analysis based on previous literature and knowledge around the impacts of phone use on sleep, cognition, and well-being.

8) Line 279/280 – you state that the relationship between bedtime phone use and sleep quality is bi-directional and complex however you have not provided any evidence that this is the case.

9) Section 4.1 Please also indicate that you did not measure mood, or record any diagnoses of sleep disorders, medical disorders, or psychiatric disorders
